# The Design of Concrete Beams Reinforced with GFRP Bars Based on Crack Width

**DOI:** 10.3390/ma15186467

**Published:** 2022-09-17

**Authors:** Jianwei Tu, Quan Zhao, Kui Gao

**Affiliations:** 1School of Civil Engineering and Architecture, Wuhan University of Technology, Wuhan 430070, China; 2State Key Laboratory of Silicate Materials for Architectures, Wuhan University of Technology, Wuhan 430070, China; 3School of Civil Engineering, Nanyang Institute of Technology, Nanyang 473000, China

**Keywords:** GFRP bars, GFRP-RC beam, flexure crack width, concrete compressive strain, design method, reinforcement ratio

## Abstract

Since glass fiber-reinforced polymer (GFRP) bars have a lower modulus than steel bars, the design of GFRP-reinforced concrete (GFRP-RC) is often governed by the serviceability limit state (deflection and cracking) rather than the ultimate state. A new design method has been proposed in this paper for GFRP-RC beams based on the flexure crack width. The state when the maximum flexure crack width in the tensile zone reaches the limit of 0.5 mm specified by ACI 440.1R-15 was used as the design limit state. The concrete compressive strain at the extreme compression fiber of concrete under the design limit state was obtained by four-point bending tests of eight full-scale GFRP-RC beams and finite element analysis. Based on the concrete compressive strain under the design limit state and cross-sectional analysis, a design method for calculating the longitudinal reinforcement ratio of GFRP-RC beams under the design limit state is proposed. This design method is proven to be feasible by the experimental and the finite element results. In addition, the flexural capacity coefficient was discussed to investigate the safety reserve of the design method.

## 1. Introduction

Fiber-reinforced polymer (FRP) bars are characterized by strong corrosion resistance, high tensile strength, light weight, and low modulus of elasticity. In the marine environment with higher humidity and saline mist, FRP bars can be adopted to replace corrosion-prone steel rebars to overcome the corrosion issue. According to fiber types, FRP bars can be classified into glass fiber-reinforced polymer (GFRP) bars, basalt fiber-reinforced polymer (BFRP) bars, carbon fiber-reinforced polymer (CFRP) bars, aramid fiber-reinforced polymer (AFRP) bars, etc. [1,2,3,4]. Compared with other FRP bars, glass FRP (GFRP) bars are widely used in practical engineering due to their inexpensive price and easy processability. Being an anisotropic material, the tensile strength of GFRP bars is higher than their compressive strength, thus making it the best tensile material in flexural members. So far, the full-scale use of GFRP bars has mostly been limited to bridge deck slabs. GFRP-RC structures, such as joints and shear walls [5,6], are only in the experimental stage. However, the product still has to find acceptance by designers because no systematic flexural member design methodology has been developed exclusively for GFRP-RC structures.

Current design methodology for GFRP-RC flexural members is based on the design method for conventional steel-RC flexural members only with some modifications and adjustments to incorporate GFRP bar-related parameters [7,8,9,10,11,12,13,14,15,16,17]. To calculate the deflection of GFRP-RC beams, the design code for steel-RC structures is consulted, and the moment of inertia of the concrete cross section is revised to be multiplied with a reduced coefficient of rigidity [12,13,14,15,16,17]. In calculating crack width, the relative bonding characteristic coefficient between GFRP bars and concrete is incorporated into the crack width formula. However, the value of the relative bonding characteristic coefficient of GFRP bars remains undefined, to be determined empirically [13]. In calculating flexural strength, the strength reduction factor for the resistance moment is amended to improve structure safety [13]. Although some studies [18,19] are trying to use crack width control for the design of GFRP-RC members, the design methods all adopt some constructional measures, such as controlling the diameter of reinforcement, the spacing of the reinforcement, and the thickness of the concrete cover, to prevent the crack width of GFRP-RC members from exceeding the specification limit under the service load, without using crack width control design from the perspective of accurate calculation. Because there is no accurate calculation of the crack width, the design method using structural measures is usually empirical and uneconomical. Therefore, it is very necessary to propose a targeted design method of GFRP-RC members based on the crack width.

Moreover, the elastic modulus of GFRP bars is smaller than that of steel bars, which makes the actual mechanical performance of GFRP-RC beams very different from steel-RC beams. When the flexure crack width of GFRP-RC beams reaches the limiting value, the stress level of GFRP bars is still far lower than its design strength; in the same loading condition, the ratio between the midspan deflection of GFRP-RC beams and that of steel-RC beams with the same rebar arrangement is about 3.0; the flexure crack width of GFRP-RC beams is usually more than three times as much as that of steel-RC beams with the same rebar arrangement [20,21,22,23]. Those studies indicate that when the GFRP bars in the beam reach the design strength, the flexure crack width has already exceeded the limit value specified by some standards [12,13,14,15,16,17]. While safety is not to be a concern here, GFRP-RC beams cannot meet the serviceability criteria because their flexure crack width and deflection are visually unacceptable.

For the above-mentioned reasons, the design limit state of GFRP-RC beams should be controlled by the flexure crack width rather than the ultimate load. Therefore, the paper proposes a design method in which the flexure crack width determines the design limit state of GFRP-RC beams. The design codes ACI 440.1R-15 [24] and CAN/CSA S6-06 [15] point out that the maximum flexure crack width of GFRP-RC beams should not exceed 0.5 mm for members subjected to aggressive environments. Therefore, the flexure crack width limiting value of GFRP-RC beams in this paper was defined as 0.5 mm. The section stress–strain relationship under this state is obtained from plane cross-section assumption, and the reinforcement ratio of tensile GFRP bars in the beam is deduced through section stress balancing condition. 

## 2. The Concrete Compressive Strain at the Extreme Compression Fiber of Concrete under the Design Limit State

Because the elastic modulus of GFRP reinforcement is small, the flexure crack width of GFRP-RC beams under loading easily exceeds the limit of the specification. Consequently, the design limit state of GFRP-RC beams should be the state when the maximum flexure crack width reaches the limit of the specification.

It is shown that when the flexure crack width reaches the limit value *W*_lim_, the concrete compressive stress of GFRP bars at the extreme compression fiber of concrete (abbreviation: concrete compressive strain) is *ε*_cd_, and the corresponding tensile strain of GFRP bars is *ε*_fd_, as shown in Figure 1. According to the plane cross-section assumption, in designing GFRP-RC beams and determining the reinforcement ratio, it is necessary to know in advance the equivalent rectangular stress block coefficient of the concrete in the compressive zone, the depth of concrete compressive zone, and the tensile strain of GFRP bars, which are all related to the concrete compressive strain. Hence, how to determine the concrete compressive strain *ε*_cd_ under the design limit state is an important topic of research for GFRP-RC beam design. Experimental research and finite element analysis were conducted on eight full-scale GFRP-RC beams, considering the influence on the concrete compressive strain exerted by those factors, such as longitudinal reinforcement ratio, concrete strength grade, and longitudinal reinforcement arrangement.

## 3. Experimental Program

### 3.1. Specimens

Eight full-scale GFRP-RC beams were tested under four-point bending until failure. All the specimens were 3000 mm long having a rectangular cross-section, which was 200 mm wide and 300 mm high. Both the longitudinal bars and stirrups were of GFRP bars. The entire beams were reinforced in compression (top) with two 12 mm diameter GFRP bars. The assembled GFRP cages are shown in Figure 2. In general, in order to prevent GFRP-RC beams from rupture of reinforcement, the reinforcement ratio of the beam should be greater than the balanced reinforcement ratio, *ρ*_fb_, where strains in concrete and GFRP bars simultaneously reach their maximum values. This is the common design concept for FRP-RC sections [1,12,15]. In this study, the reinforcement ratio of all beams exceeded the balanced reinforcement ratio.

Figure 3 shows the detailed dimensions and loading position of specimens. The specimen numbering scheme for this experiment was Ga-b-c-d, where G represents GFRP bar; a represents the concrete strength, including C30 and C50; b represents the reinforcement ratio in the tensile zone; c represents the longitudinal bar arrangement in the tensile zone, with 1 meaning one layer and 2 meaning two layers (clear spacing between two layers is 25 mm); d represents the stirrups at the pure bending section of the span, with 0 meaning no stirrups and 1 meaning having stirrups. The details of the specimens are specified in Table 1.

### 3.2. Materials

Table 2 shows the mechanical properties of GFRP bars used in the test. In order to analyze the influence of concrete strength on the bond behavior, two different concrete strengths were used. Three concrete cubes of 150 mm × 150 mm × 150 mm were reserved and then cured for 28 days with all specimens being subjected to the same conditions, obtaining an average compressive strength of 30.7 ± 0.5 MPa and 44.9 ± 0.7 MPa.

### 3.3. Instrumentation and Testing

All beams were tested under four-point bending over a simply supported clear span of 3000 mm. The distributive girders were used to apply symmetric and simultaneous load at two loading points, forming a pure bending section in the midspan of the test beam. Five linear variable differential transducers (LVDTs) were installed on each beam to measure deflection at different locations during testing. The crack width gauge was used to measure the flexure crack width. The reinforcing bars and the compression concrete zone of the beams were instrumented with electrical resistance strain gauges to capture the strains at the desired locations. Figure 4 shows a beam specimen during testing.

The load was applied in loading control at 2 kN/min until the specimens were about to crack. After the initiation of cracking, the loading rate was decreased to 1 kN/min and kept for 2 min to record the experimental data in the stable condition of load and deformation.

## 4. Finite Element 

Due to the small number of specimens, the test results have a certain degree of discreteness. In this paper, accurate finite element modeling was used to reproduce the crack development process of the specimens. This can ensure the effectiveness of the design method of GFRP-RC beams proposed in this paper.

### 4.1. Material Model 

In the linear elastic stage, the behavior of concrete was determined using the elastic modulus *E*_c_ and Poisson’s ratio (0.2). In the plastic stage, the concrete damage plastic model (CDP) was used to simulate the inelastic behavior of concrete. The nonlinear concrete behavior was calculated by using uniaxial compression/tensile stress–strain data and the following parameters. According to Refs. [25,26], the yield surface was determined with the ratio of initial biaxial compressive yield stress to initial uniaxial compressive yield stress (1.16) and the ratio of the second stress invariant on the tensile meridian to that on the compressive meridian (0.667). The dilation angle (35°) and the flow potential eccentricity (0.1) modified the nonassociated potential flow. The viscosity parameter (0.0001) was used for the viscoplastic regularization of the concrete constitutive equation. According to the Chinese code (GB50010-2010) [27], the compressive stress (*σ*_c_)–compressive strain (*ε*_c_) relationship of concrete is expressed as:(1)$$\{\begin{array}{cc} \sigma_{c}=f_{c}\left[1-{\left(1-\frac{\varepsilon_{c}}{\varepsilon_{0}}\right)}^{2}\right] & \varepsilon_{c}\leq \varepsilon_{0}\\ \sigma =f_{c} & \varepsilon_{0}<\varepsilon_{c}\leq \varepsilon_{\mathrm{cu}} \end{array}$$

The tensile stress (*σ*_t_)–tensile strain (*ε*_t_) relationship of concrete is expressed as:(2)$$\sigma_{t}=f_{t,r}\frac{\varepsilon_{t}/\varepsilon_{t,r}}{{\left(\varepsilon_{t}/\varepsilon_{t,r}-1\right)}^{1.7}+\varepsilon_{t}/\varepsilon_{t,r}}$$
where *f*_c_ is the concrete compressive strength; *ε*_0_ is the concrete compressive strain corresponding to the concrete compressive strength; *ε*_cu_ is the concrete compressive strain under the ultimate limit state, *ε*_tr_ is the tensile cracking strain.

GFRP bars were simulated as elastic brittle materials considering a Poisson’s ratio of 0.2.

### 4.2. Bond–Slip Model

In order to simulate the bond–slip behavior between GFRP bars and concrete, spring element connection was used in this study. Several groups of three-dimensional spring elements were evenly distributed along the FRP bars on the side opposite of the load. In each group, the two spring elements perpendicular to the specimen height were set as linear spring elements, while the elements along the specimen height were set as nonlinear spring elements. In earlier pull-out tests [27] on the GFRP bar used in this study, the bond stress–slip relationship (as shown Figure 5) was obtained and used to describe the mechanical properties of the nonlinear spring elements.

### 4.3. Cracking Model

The extended finite element method (XFEM) has developed to become a standard tool in fracture mechanics with a sharp crack representation and can well describe the propagation of macrocracks [28,29]. In this paper, the XFEM module in ABAQUS [30] was used to carry out the extended finite element calculation. The maximum principal stress cracking criterion and the linear softening damage model based on energy were used to simulate concrete. The initial crack spacing preset by the model was the same as the experimental results.

### 4.4. Boundary, Element, and Mesh

A three-dimensional FE model for FRP-RC beams was developed in ABAQUS. The geometry, loading mode, and boundary conditions were similar to those used in the experiments. The FE model is shown in Figure 6. Herein, a three-dimensional 8-node reduced integral (C3D8R) element was used to simulate the concrete, while the internal reinforcement bars were modeled using three-dimensional 2-node linear truss (T3D2) elements, and the same mesh size as that was used for concrete. To ensure convergence and minimize the computational time, a mesh sensitivity analysis was conducted. In this work, a mesh size of 12.5 mm was used to present the cracking patterns as shown.

### 4.5. Verification of the FEM Results

Figure 7 shows a comparison between the FEM and experimental results of GFRP-RC beams. It is clear that the initial stiffness and the maximum load calculated with the FEM are higher than those of the experimental results. However, the overall trends and key features of the simulated load–deflection curves are consistent with those obtained through experimental results.

## 5. Results and Analysis

At each load stage, the flexure crack width and the corresponding concrete compressive strain can be measured. The flexure crack width measured is at the height of the reinforcement. When the flexure crack width reaches the limit of 0.5 mm, this state is determined to be the design limit state of GFRP-RC beams. By averaging the concrete compressive strain measured by the two strain gauges, the concrete compressive strain is obtained under the design limit state, and its finite element calculation value is obtained as well. 

Figure 8 shows the overview of the specimens after failure. Since the reinforcement ratio of all specimens exceeds the balanced reinforcement ratio, concrete crushing failure occurs in all specimens, as shown in Figure 8. The crack patterns of the tested beams at 0.30 *M*_n_ (*M*_n_ is the nominal moment of the reinforced-concrete section), as shown in Figure 9, reveal that increasing the reinforcement ratio increased the number of cracks and, consequently, reduced the average crack spacing. The compressive concrete strength and the longitudinal bar arrangement have no obvious effect on the number and spacing of the cracks. Figure 9 also shows that the development of cracks is very close to the position of the stirrups. 

Figure 10 indicates the relationship between the concrete compressive strain under the design limit state and the longitudinal reinforcement ratio and concrete strength. It is shown that when the maximum flexure crack width reaches the limit of 0.5 mm, the concrete compression strain changes just a little with the increase in the longitudinal reinforcement ratio. The results of both the experimental and the finite element analysis show that the concrete strength almost has no influence on the concrete compressive strain value under the design limit state. 

Figure 11 indicates the relationships between the applied load and the flexure crack width. The first crack appeared at the cracking load level. For the GFRP-RC beams, the crack width varied linearly with the applied moment until failure. A slight reduction rate of crack width was observed for some beams, but this can be considered linear behavior. Clearly, crack width is influenced by the stirrups. After the crack width reaches 0.5 mm, the crack width of specimen G30-1.1-2-0 increases faster than specimen G30-1.1-2-1.

Figure 12 indicates the relationships between the applied load and the concrete compression strain. It is shown that the concrete compressive strain is less influenced by the influence of stirrups and more influenced by the longitudinal bar arrangement. After the concrete compressive strain reaches 0.001, the concrete compressive strain of specimen G30-1.1-1-1 increases faster than specimen G30-1.1-2-1.

Figure 13 reflects the relationship between the concrete compressive strain under the design limit state and the longitudinal reinforcement arrangement. It is evident that when the flexure crack width reaches the limit value of 0.5 mm, the concrete compressive strain becomes slightly increased by the identical reinforcement ratio arranged in two layers than by that arranged in one layer and that the concrete compressive strain becomes higher when there is no stirrup arrangement than when stirrups are present. Overall, the reinforcement arrangement has very little effect on the concrete compressive strain. 

Figure 14 indicates the relationships between the reinforcement ratio and the concrete compressive strain under the design limit state. It is indicated that the experimental compressive strain values under the design limit state range from 0.000943 to 0.001176, and the average value, mean square error, and coefficient of variation are 0.00107, 0.00009, and 0.084, respectively; the finite element calculation values of the compressive strain under the design limit state range from 0.000935 to 0.001103 and the average value, mean square error, and coefficient of variation are 0.000989, 0.000055, and 0.056, respectively. 

From the above results, under the design limit state, the concrete compressive strain of each beam obtained by test and finite element analyses is close to 0.001, and it is virtually not influenced by the factors, such as reinforcement ratio, concrete strength, reinforcement arrangement, etc. Thus, the concrete compressive strain under the design limit state can be naturally set to be 0.001, which provides a reliable basis for the further design of GFRP-RC beams.

## 6. Design of GFRP-RC Beams

### 6.1. Equivalent Rectangular Stress Block Coefficient under the Design Limit State 

Figure 15 shows the strain and the stress distributions in the cross section under the design limit state. In Figure 15, *h* is the height of beam section; *h*_0f_ is the effective height of the beam section; *x*_cd_ is the height of the concrete compression zone under the design limit state; *ε*_cd_ and *f*_cd_ are the concrete compressive strain and stress under the design limit state, respectively; *ε*_fd_ and *f*_fd_ are the tensile strain and stress of the GFRP bar under the design limit state, respectively; *α*_d_ and *β*_d_ are the strain-dependent parameter and stress-dependent parameter under the design limit state, respectively. As can be seen from Figure 15, the concrete compressive strain at the extreme compression fiber of concrete, *ε*_cd_, does not reach the ultimate concrete compressive strain *ε*_cu_, under the design limit state. The stress distribution in the concrete can be approximated with an equivalent rectangular stress block using two parameters, *α*_d_ and *β*_d_, as shown in Figure 15. To compute the equivalent stress block parameters, *α*_d_ and *β*_d_, the stress–strain relationship in concrete needs to be determined. In this study, the concrete stress–strain relationship was determined by the current Chinese Code for Design of Concrete Structures GB50010-2010 [26], as shown in Equation (1).

In Figure 15, when the flexure crack width reached the limit of 0.5 mm, the resultant force *F* and resultant force moment *M* of the beam section in the compressive zone can be depicted as: (3)$$F=b\int_{0}^{x_{\mathrm{cd}}}f_{c}\left[1-{\left(1-\frac{\varepsilon \left(x\right)}{\varepsilon_{0}}\right)}^{2}\right]dx$$
(4)$$M=b\int_{0}^{x_{\mathrm{cd}}}f_{c}\left[1-{\left(1-\frac{\varepsilon \left(x\right)}{\varepsilon_{0}}\right)}^{2}\right]\left(x_{\mathrm{cd}}-x\right)dx$$

According to plane cross-section assumption, providing
(5)$$\frac{x}{\varepsilon }=\frac{x_{\mathrm{cd}}}{\varepsilon_{\mathrm{cd}}}=k$$
it is obtained that
(6)$$F=kf_{c}b\int_{0}^{\varepsilon_{\mathrm{cd}}}\left[1-{\left(1-\frac{\varepsilon }{\varepsilon_{0}}\right)}^{2}\right]d\varepsilon =kf_{c}b\varepsilon_{\mathrm{cd}}\left[\frac{\varepsilon_{\mathrm{cd}}}{\varepsilon_{0}}-\frac{1}{3}{\left(\frac{\varepsilon_{\mathrm{cd}}}{\varepsilon_{0}}\right)}^{2}\right]$$
(7)$$M=k^{2}f_{c}b\int_{0}^{\varepsilon_{\mathrm{cd}}}\left[1-{\left(1-\frac{\varepsilon }{\varepsilon_{0}}\right)}^{2}\right]\left(\varepsilon_{\mathrm{cd}}-\varepsilon \right)d\varepsilon =k^{2}f_{c}b\varepsilon_{\mathrm{cd}}^{2}\left[\frac{1}{3}\left(\frac{\varepsilon_{\mathrm{cd}}}{\varepsilon_{0}}\right)-\frac{1}{12}{\left(\frac{\varepsilon_{\mathrm{cd}}}{\varepsilon_{0}}\right)}^{2}\right]$$

Then, the block parameters *α*_d_ and *β*_d_ can be calculate as:(8)$$\{\begin{array}{c} \beta_{d}=\frac{2M}{Fx_{\mathrm{cd}}}=\frac{2M}{Fk\varepsilon_{\mathrm{cd}}}\\ \alpha_{d}=\frac{F}{f_{c}b\beta_{d}x_{\mathrm{cd}}}=\frac{F}{f_{c}b\beta_{d}k\varepsilon_{\mathrm{cd}}} \end{array}$$

By substituting Equations (6) and (7) into Equation (8), it is obtained that
(9)$$\{\begin{array}{c} \beta_{d}=\frac{4-\frac{\varepsilon_{\mathrm{cd}}}{\varepsilon_{0}}}{6-2\frac{\varepsilon_{\mathrm{cd}}}{\varepsilon_{0}}}\\ \alpha_{d}=\frac{1}{\beta }\left[\frac{\varepsilon_{\mathrm{cd}}}{\varepsilon_{0}}-\frac{1}{3}{\left(\frac{\varepsilon_{\mathrm{cd}}}{\varepsilon_{0}}\right)}^{2}\right] \end{array}$$

In the second section of Figure 15, it is elaborated that under the design limit state the concrete compressive strain is 0.001 and the peak concrete compressive strain in Equation (1) is set to be 0.002 [26], which are substituted into Equation (9), obtaining *α*_d_ = 0.6 and *β*_d_ = 0.7.

### 6.2. The Longitudinal Reinforcement Ratio in the Tensile Zone

It is shown in Figure 15 that the equilibrium condition of force and bending moments can lead to:(10)$$\alpha_{d}f_{c}b\beta_{d}x_{\mathrm{cd}}=E_{f}\varepsilon_{\mathrm{fd}}A_{f}$$
(11)$$M_{d}=E_{f}\varepsilon_{\mathrm{fd}}A_{f}\left(h_{0f}-0.5\beta_{d}x_{\mathrm{cd}}\right)$$
where *M*_d_ is the bending moment under the design limit state.

The depth of the compressive zone can be obtained by the plane cross-section strain relation:(12)$$x_{\mathrm{cd}}=\frac{\varepsilon_{\mathrm{cd}}}{\varepsilon_{\mathrm{fd}}+\varepsilon_{\mathrm{cd}}}h_{0f}$$

Substitute Equation (10) into Equation (11), obtaining
(13)$$x_{\mathrm{cd}}=\frac{1}{\beta_{d}}\left(h_{0f}-\sqrt{h_{0f}^{2}-\frac{2M_{d}}{\alpha_{d}f_{c}b}}\right)$$

Combine Equations (12) and (13), obtaining
(14)$$\varepsilon_{\mathrm{fd}}=\left(\frac{h_{0f}}{x_{\mathrm{cd}}}-1\right)\varepsilon_{\mathrm{cd}}$$

Combine Equations (10) and (14), obtaining the longitudinal reinforcement ratio *ρ*_fd_ of the GFRP bar in the tensile zone under the design limit state
(15)$$\rho_{\mathrm{fd}}=\frac{\alpha_{d}\beta_{d}f_{c}x_{\mathrm{cd}}}{E_{f}\varepsilon_{\mathrm{fd}}h_{0f}}$$

So far, the longitudinal reinforcement ratio can be determined in the GFRP-RC beam with a rectangular section.

### 6.3. The Ultimate Limit State

The ultimate limit state is that the failure of the GFRP-RC beam is initiated by the crushing of the concrete and the strain, and the stress distribution in the cross section in the concrete can be described by Figure 16. In Figure 16, *x*_cu_ is the height of the concrete compression zone under the ultimate limit state; *ε*_cu_ and *f*_cu_ are the concrete compressive strain and stress under the ultimate limit state, respectively; *ε*_fl_ and *f*_fl_ are the tensile strain and stress of the GFRP bar under the ultimate limit state, respectively; *α*_u_ and *β*_u_ are the strain-dependent parameter and stress-dependent parameter under the ultimate limit state, respectively.

The concrete strain at the maximum compressive concrete strength is assumed to be 0.0033 according to GB50010-2010 [27]. The coefficient of equivalent rectangular stress block can be calculated with Equation (16) [10]:(16)$$\{\begin{array}{c} \beta_{u}=\frac{1-\frac{2}{3}\frac{\varepsilon_{0}}{\varepsilon_{\mathrm{cu}}}+\frac{1}{6}{\left(\frac{\varepsilon_{0}}{\varepsilon_{\mathrm{cu}}}\right)}^{2}}{1-\frac{\varepsilon_{0}}{3\varepsilon_{\mathrm{cu}}}}\\ \alpha_{u}=\frac{1}{\beta_{u}}\left(1-\frac{\varepsilon_{0}}{3\varepsilon_{\mathrm{cu}}}\right) \end{array}$$

Additionally, *α*_u_ = 0.97, *β*_u_ = 0.82.

Similarly, according to the force equilibrium condition, we substituted *A*_f_ = *ρ*_f_*bh*_0f_ into Equation (10), obtaining
(17)$$\alpha_{u}\beta_{u}f_{c}\frac{\varepsilon_{\mathrm{cu}}}{\varepsilon_{\mathrm{fl}}+\varepsilon_{\mathrm{cu}}}=E_{f}\varepsilon_{\mathrm{fI}}\rho_{\mathrm{fb}}$$
namely,
(18)$$\varepsilon_{\mathrm{fI}}=\frac{1}{2}\left(\sqrt{\varepsilon_{\mathrm{cu}}^{2}+4\lambda }-\varepsilon_{\mathrm{cu}}\right)$$
in which *λ* = *α*_u_*β*_u_
*f*_c_*ε*_cu_/(*E*_f_*ρ*_fb_).

So, the ultimate bending moment of the GFRP-RC beam is:(19)$$M_{u}=E_{f}\varepsilon_{\mathrm{fI}}\rho_{\mathrm{fb}}bh_{0f}^{2}\left(1-\frac{0.5\beta_{u}\varepsilon_{\mathrm{cu}}}{\varepsilon_{\mathrm{fI}}+\varepsilon_{\mathrm{cu}}}\right)$$

## 7. The Comparison between Theoretical Calculations and Experimental Results

### 7.1. The Design Limit State

Table 3 lists the experimental crack width, the applied load, the experimental bending moment, the theoretical bending moment, and the FEM bending moment under the design limit state. Table 3 illustrates that when the concrete compressive strain is close to 0.001, the maximum flexure crack width is normally between 0.3 and 0.52 mm, which explains further that it is appropriate to determine the concrete compressive strain to be 0.001 in designing GFRP-RC beams so that the maximum flexure crack width is no more than the limiting value of 0.5 mm. The average and COV of *M*_d,ex_/*M*_d,th_ are 1.06 and 0.08, respectively. The average and COV of *M*_d,ex_/*M*_d,fe_ are 1.07 and 0.05, respectively. The theoretical and FEM bending moments are in good agreement with the experimental bending moment, meaning the calculation method in this paper is feasible.

### 7.2. The Ultimate Limit State

Table 4 lists the concrete compressive strain, the experimental maximum flexure crack width, the applied load, the experimental bending moment, the theoretical bending moment, and the FEM bending moment under the ultimate limit state. It can be seen from Table 4 that the average of *M*_u,ex_/*M*_u,th_ and *M*_u,ex_/*M*_u,fe_ is close to 1 and the COV is smaller, which proves that the theoretical calculation method and the finite element method are effective.

## 8. Flexural Capacity Coefficient

The flexural capacity coefficient (*S*_J_) of GFRP-RC beams can be represented as [31]:(20)$$S_{J}=\frac{M_{u}}{M_{0.001}}$$

Figure 17 compares the flexural capacity coefficient of experimental, theoretical, and finite elements, showing that when the reinforcement ratio is greater than the balanced reinforcement ratio, the flexural capacity coefficient of the GFRP-RC beam is always about 2.5, proving the high factor of safety of GFRP-RC beams from the design limit state to the ultimate limit state. With regard to the GFRP bar, creep rupture will take place if it experiences too much stress when in service. If the stress of the GFRP bar is controlled within 30% of the ultimate strength, its creep rupture can be avoided [32]. With the design method proposed in this paper, generally the strain of the GFRP bar is no greater than 0.004, and the stress is no more than 180 MPa, which is within 30% of the ultimate strength. This stress does not produce creep rupture of the GFRP bar, maintaining its strong flexural capacity even after long service periods. 

## 9. Conclusions

This paper mentions the derivation of formulas for the calculation of the longitudinal reinforcement ratio while setting the moment when the flexure crack width reaches the limiting value of 0.5 mm as the design limit state of GFRP-RC beams. It also describes the experimental and finite element analyses of eight GFRP beams, drawing the conclusion as follows:(1)If the longitudinal reinforcement ratio is greater than the balanced reinforcement ratio, when the flexure crack width of the GFRP-RC beam reaches the standard limit value of 0.5 mm, the concrete compressive strain at the extreme compression fiber of the concrete is about 0.001. The longitudinal reinforcement ratio, concrete strength, and longitudinal reinforcement arrangement have little influence on the concrete compressive strain of this state.(2)Great changes take place in the stress–strain relation of GFRP-RC beams under the design limit state and the ultimate limit state. By setting the concrete compressive strain at the extreme compression fiber of concrete as 0.001, the equivalent rectangular stress block coefficient, the depth of the concrete compressive zone, and the strain of the tensile bar are calculated again. Finally, a formula for the calculation of the longitudinal reinforcement ratio under this state was derived.(3)The theoretical bending moment under the design limit state was compared with its experimental value. The ratio between them ranges from 0.92 to 1.04. Their good agreement proves that the calculation in this paper is correct.(4)When the concrete of the test beams in the compressive zone is crushed under the ultimate limit state, the concrete crack width in the tensile zone reaches 1.2–2.0 mm, far beyond the standard limiting value of 0.5 mm, which explains further that it is appropriate to treat the moment when the concrete crack width in the tensile zone reaches the limiting value as the design limit state, reasonably reflecting the property of the GFRP bar and the actual working conditions of the beam. Furthermore, the ratio between the theoretical and experimental flexural bending moments under the ultimate limit state ranges from 0.85 to 0.96, and hence it is in a good agreement.(5)GFRP-RC beams based on the design method proposed in this paper have the flexural capacity coefficient of about 2.5, which means the beams have a very high factor of safety and reserve strength. Generally, the stress of the tensile bar material is no more than 30% of the ultimate strength, which makes it safe against creep failure too, thereby ensuring its long-term performance.

## Figures and Tables

**Figure 1 materials-15-06467-f001:**
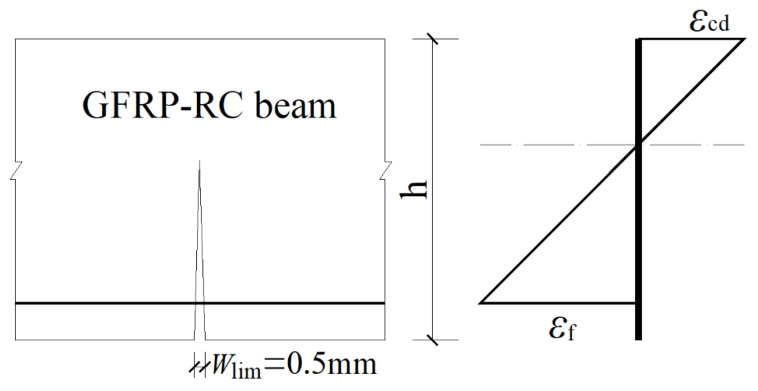
The section strain distribution of GFRP-RC beam under the design limit state.

**Figure 2 materials-15-06467-f002:**
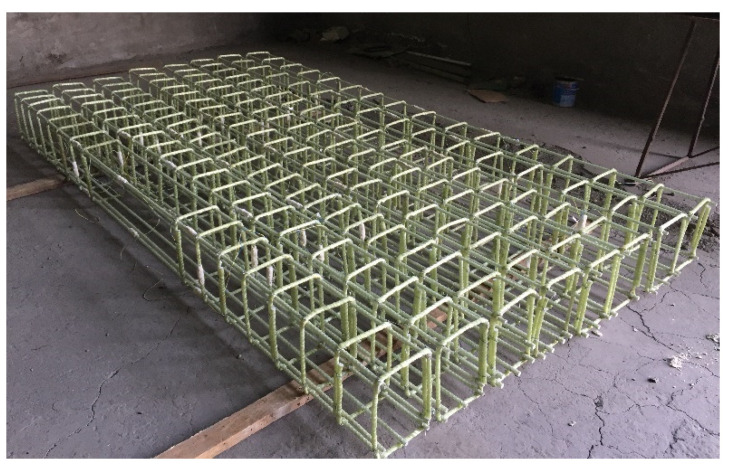
Overview of the assembled GFRP cages.

**Figure 3 materials-15-06467-f003:**
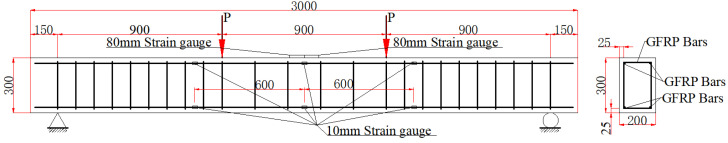
Detailed dimensions and loading position of specimens.

**Figure 4 materials-15-06467-f004:**
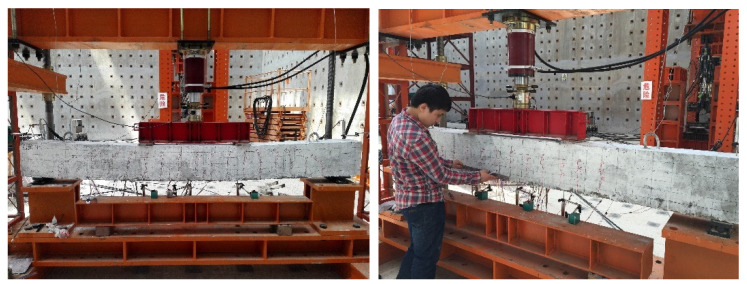
Test specimen and setup.

**Figure 5 materials-15-06467-f005:**
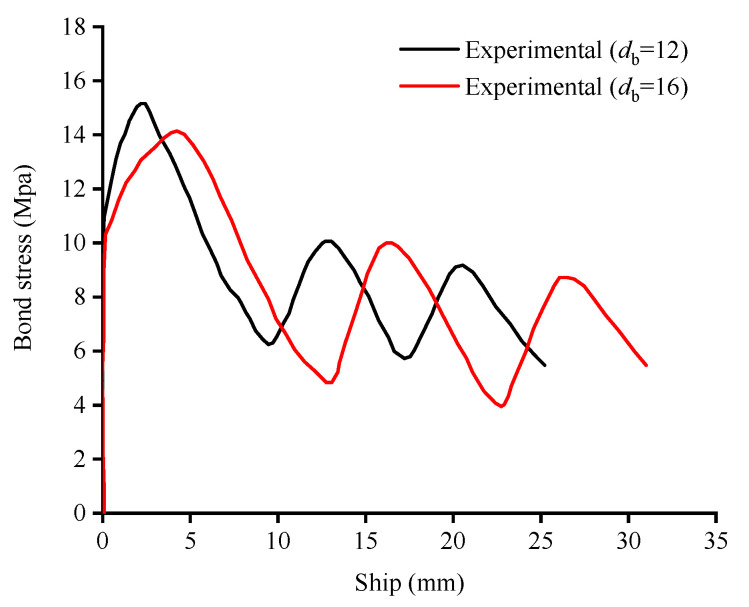
Bond stress–slip relationship of GFRP bars.

**Figure 6 materials-15-06467-f006:**
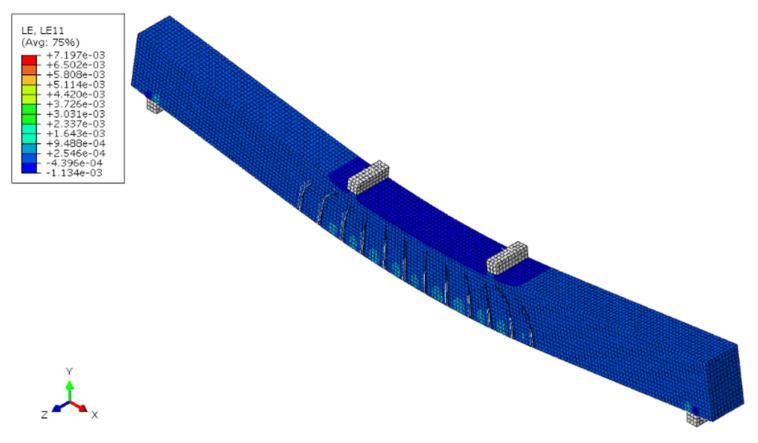
The finite element analysis model of GFRP-RC beam.

**Figure 7 materials-15-06467-f007:**
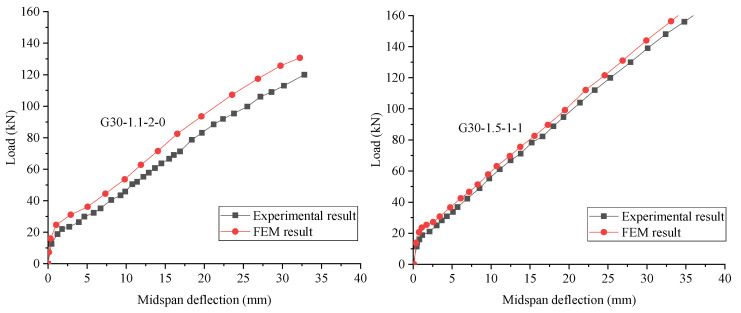
Comparison of FEM and experimental results.

**Figure 8 materials-15-06467-f008:**
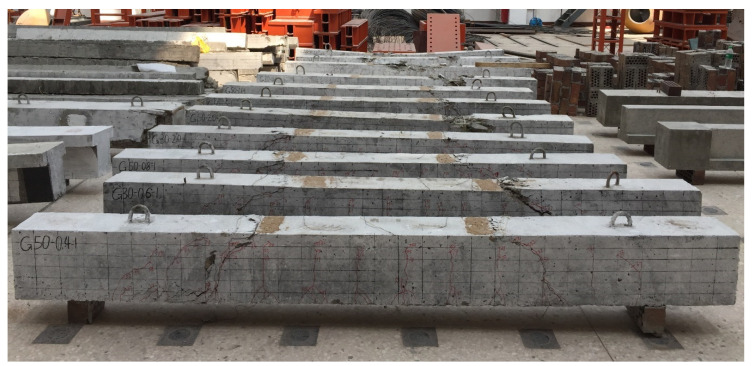
Overview of the specimens after failure.

**Figure 9 materials-15-06467-f009:**
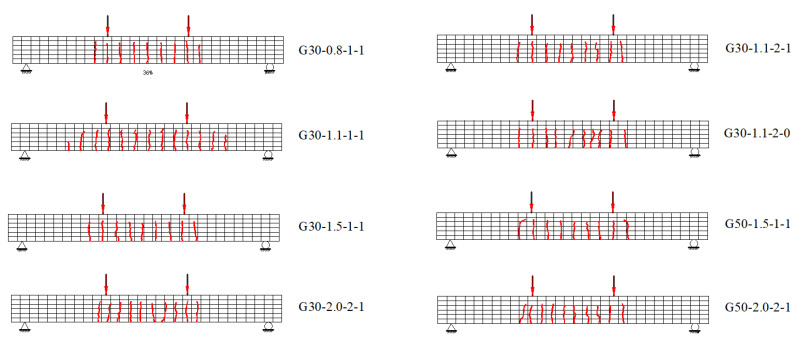
Crack patterns of specimens.

**Figure 10 materials-15-06467-f010:**
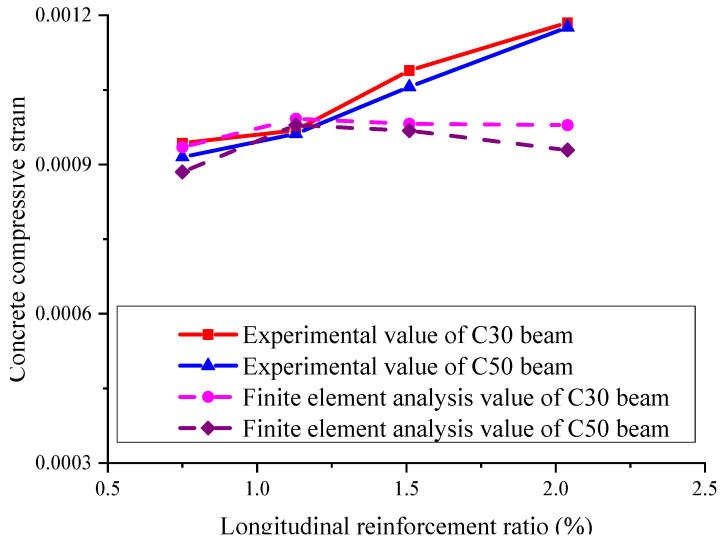
The influence of the longitudinal reinforcement ratio and concrete strength on concrete compressive strain.

**Figure 11 materials-15-06467-f011:**
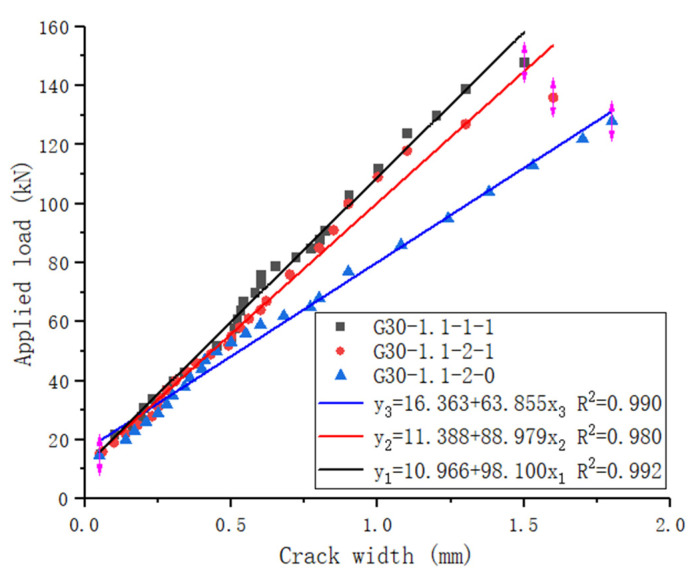
The relationships between applied load and flexure crack width.

**Figure 12 materials-15-06467-f012:**
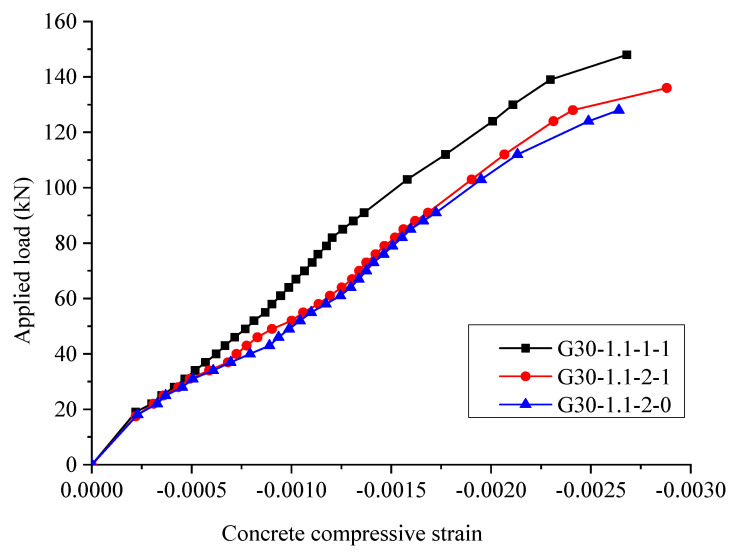
The relationships between applied load and the concrete compressive strain.

**Figure 13 materials-15-06467-f013:**
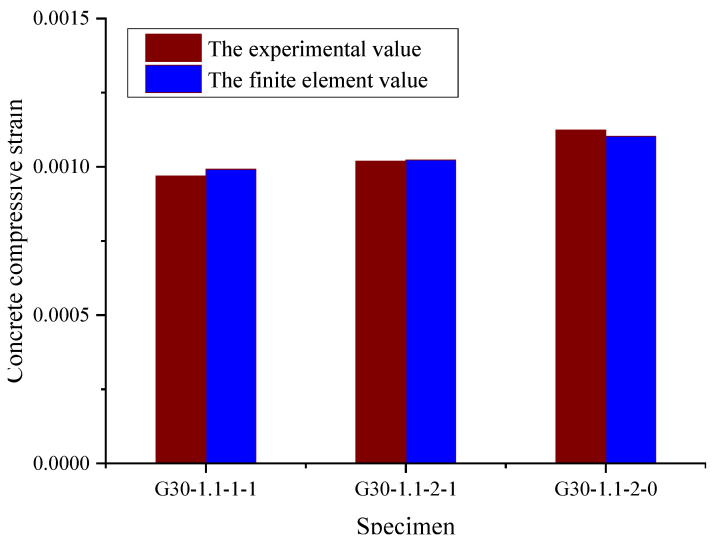
The influence of longitudinal reinforcement and stirrup arrangement on concrete compressive strain.

**Figure 14 materials-15-06467-f014:**
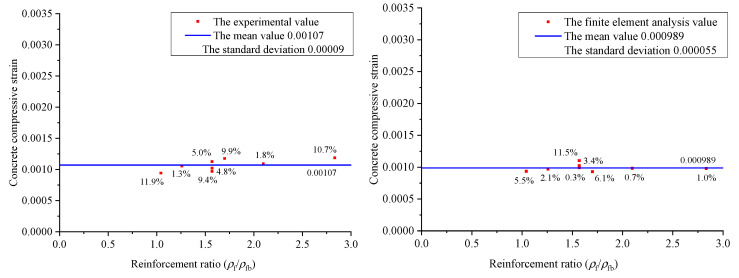
The relationships between reinforcement ratio and the concrete compressive strain under the design limit state.

**Figure 15 materials-15-06467-f015:**
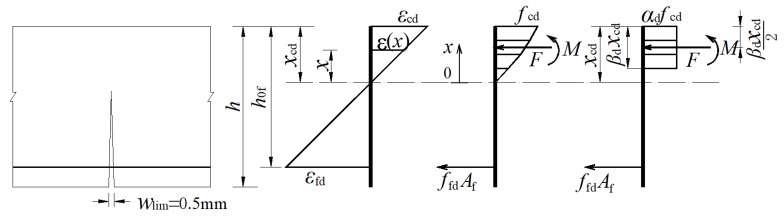
The section stress and strain distributions under the design limit state.

**Figure 16 materials-15-06467-f016:**
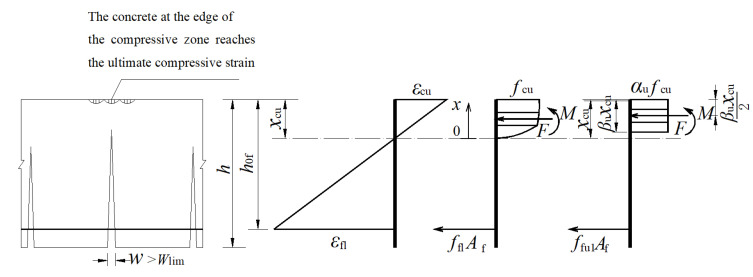
The section stress and strain distributions under the ultimate limit state.

**Figure 17 materials-15-06467-f017:**
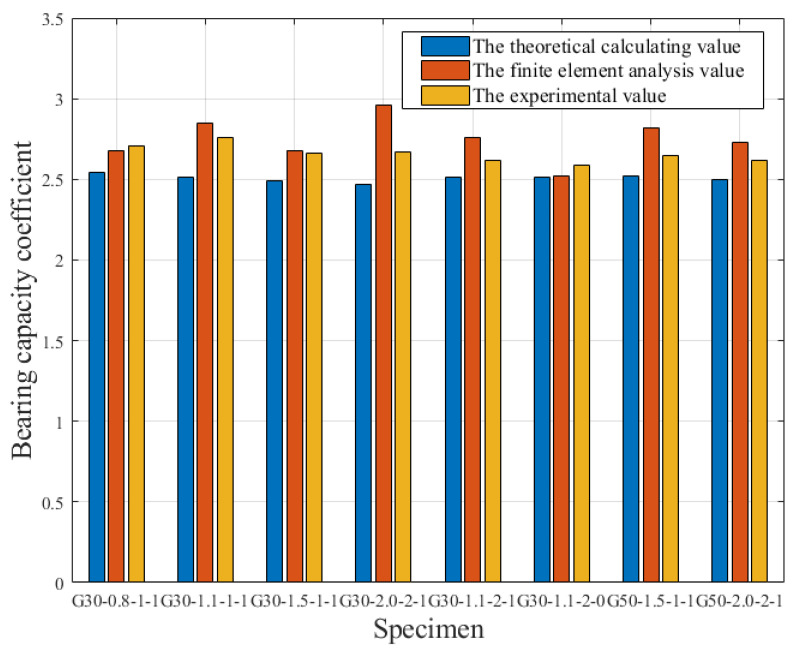
The comparison between the theoretical, experimental, and finite element flexural capacity coefficients.

**Table 1 materials-15-06467-t001:** Details of the test specimens.

Specimen	*f* _c_	*ρ*_fb_ (%)	Longitudinal Reinforcement	*ρ*_f_(%)	Stirrups (Two Ends/Pure Bending Section)	*h*_0f_ (mm)
G30-0.8-1-1	C30	0.72	2φ16	0.75	φ12@150/150	267
G30-1.1-1-1	C30	0.72	3φ16	1.13	φ12@150/150	267
G30-1.5-1-1	C30	0.72	4φ16	1.51	φ12@150/150	267
G30-2.0-2-1	C30	0.72	5φ16	2.04	φ12@130/130	247
G30-1.1-2-1	C30	0.72	5φ12	1.13	φ12@150/150	251
G30-1.1-2-0	C30	0.72	5φ12	1.13	φ12@150/-	251
G50-1.5-1-1	C50	1.20	4φ16	1.51	φ12@150/150	267
G50-2.0-2-1	C50	1.20	5φ16	2.04	φ12@130/130	247

Note: *f*_c_ is concrete strength; *ρ*_fb_ is balanced reinforcement ratio; *ρ*_f_ is longitudinal reinforcement ratio of tensile GFRP bar; *h*_0f_ is effective height of beam section.

**Table 2 materials-15-06467-t002:** Properties of GFRP bar.

*d*_b_ (mm)	*f*_fu_ (MPa)	*E*_f_ (GPa)	*ε* _fu_
12	660	44.25	0.015
16	578	40.69	0.014

Notes: *d*_b_ is bar diameter; *f*_fu_ is ultimate tensile strength; *E*_f_ is modulus of elasticity in tension; *ε*_fu_ is ultimate tensile strain.

**Table 3 materials-15-06467-t003:** The comparison between the experimental and theoretical bending moments under the design limit state.

Specimen	*ε* _cd_	*W*_d,ex_ (mm)	*P*_d,ex_ (kN)	*M*_d,ex_ (kN˙m)	*M*_d,th_ (kN˙m)	*M*_d,fe_ (kN˙m)	*M*_d,ex_/*M*_d,th_	*M*_d,ex_/*M*_d,fe_
G30-0.8-1-1	0.00094	0.52	22.16	19.94	19.78	20.79	1.01	0.96
G30-1.1-1-1	0.00096	0.50	26.80	24.12	23.46	22.32	1.03	1.08
G30-1.5-1-1	0.00109	0.40	31.90	28.71	23.36	25.86	1.23	1.11
G30-2.0-2-1	0.00119	0.30	30.76	27.68	25.35	26.23	1.09	1.06
G30-1.1-2-1	0.00102	0.49	25.96	23.36	21.45	21.28	1.09	1.10
G30-1.1-2-0	0.00112	0.45	24.76	22.28	20.45	22.26	1.09	1.00
G50-1.5-1-1	0.00106	0.40	34.56	31.10	33.44	27.83	0.93	1.12
G50-2.0-2-1	0.00118	0.36	36.30	32.67	31.35	29.65	1.04	1.10
Average							1.06	1.07
Standard deviation							0.08	0.05
Coefficient of variation (COV) (%)							0.08	0.05

Note: *W*_d,ex_ is the experimental value of maximum flexure crack width; *P*_d,ex_ is the experimental value of the applied load under the design limit state; *M*_d,ex_ is the experimental bending moment under the design limit state; *M*_d,th_ is the theoretical bending moment under the design limit state, *M*_d,fe_ is the FEM bending moment under the design limit state.

**Table 4 materials-15-06467-t004:** The comparison between the experimental bending moment and the theoretical bending moment under the ultimate limit state.

Specimen	*ε* _cu_	*W*_u,ex_ (mm)	*P*_u,ex_ (kN)	*M*_u,ex_ (kN˙m)	*M*_u,th_ (kN˙m)	*M*_u,fe_ (kNx2D9;m)	*M*_u,ex_/*M*_u,th_	*M*_u,ex_/*M*_u,fe_
G30-0.8-1-1	0.0033	1.70	60.0	54.00	50.25	52.27	1.07	1.03
G30-1.1-1-1	0.0033	1.50	74.0	66.60	58.98	61.55	1.13	1.08
G30-1.5-1-1	0.0033	1.48	85.0	76.50	65.70	68.63	1.16	1.11
G30-2.0-2-1	0.0033	1.20	82.0	73.80	62.56	65.2	1.18	1.13
G30-1.1-2-1	0.0033	1.60	68.0	61.20	53.80	56.01	1.14	1.09
G30-1.1-2-0	0.0033	1.80	64.0	57.60	53.80	56.01	1.07	1.03
G50-1.5-1-1	0.0033	2.00	91.5	82.35	81.68	85.83	1.01	0.96
G50-2.0-2-1	0.0033	1.40	95.0	85.50	78.26	82.17	1.09	1.04
Average							1.11	1.06
Standard deviation							0.05	0.05
Coefficient of variation (COV) (%)							0.05	0.05

Note: *W*_u,ex_ is the experimental value of maximum flexure crack width under the ultimate limit state; *P*_u,ex_ is the applied load under the ultimate limit state; *M*_u,ex_ is the experimental bending moment under the ultimate limit state; *M*_u,th_ is the theoretical bending moment under the ultimate limit state, *M*_u,fe_ is the FEM bending moment under the ultimate limit state.

## Data Availability

Not applicable.

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
