# Peer review of "The Design of Concrete Beams Reinforced with GFRP Bars Based on Crack Width"

_materials, 2022, doi:10.3390/ma15186467_

Round 1

Reviewer 1 Report

The paper proposes a new design method for GFRP-RC beams based on the flexure crack width. It presents theoretical, numerical and experimental results and for this reason the methodology adopted is very strong. However, some changes and clarifications are required before publication.

General comment

- There are some typos in the manuscript ("base on" in the abstract, ...).

Section 2

- Fig. 1: Please improve the quality of the figure and its description. For example, add + and - in the diagram and explain why the diagram is not on the whole section. You mention something after Fig. 10.

- Fig. 2: Please improve the quality. Maybe a photo of one specimen should be helpful.

Section 3.3

-"Forming a pure bending section in the mid-span of the test beam": what do you mean?

- "An increment of 5kN is applied at every stage": Is the first stage 0 kN? How long is each stage?

Section 4

- "Two nodes are created in each element,  placed on the concrete and the GFRP bar each": A drawing would be helpful. Why do you use springs instead of contact?

- Improve the description of the FE model: boundary conditions, material law, ...

Section 5

- In my opinion, you should report in each table the discrepancy % between the theoretical and/or numerical and/or experimental results.

 - "It is shown that the crack width basically exhibits a linear variation with increasing applied load": This sentence is too strong and should be justified. How much is R^2 if you assume a linear regression model?

Section 6

- Eq. 8: What is beta?

Section 6.3

- "The concrete strain at the maximum compressive concrete strength is assumed to be 0.0033 according to GB50010-2010": A reference number is missing.

References

- The most recent references are two from 2016, two from 2017 and one from 2018. Please add more recent documents.

Reviewer 2 Report

Manuscript Number: materials-1886579

Full Title: The design of concrete beam reinforced with GFRP bars based on crack width

In this manuscript, A new design method has been proposed for GFRP-RC beams based on the flexure crack width. The authors claim that “Current design methodology for GFRP-RC flexural members is based on the design method for conventional steel-RC flexural members only with some modifications and adjustments to incorporate GFRP bar related parameters”, but this can be argued since there is a bunch of literature regarding the serviceability of such structures, e.g., see:

·         Reinforced Concrete with FRP Bars Mechanics and Design, 2014, CRC Press.

 The overall quality of the paper is acceptable, but the novelty of this research cannot be attested as claimed by the authors. The literature used for this research seems outdated, and recent research work should be explored. Major revisions of the manuscript, especially regarding the novelty and way of presentation, are needed.

Comments:  

·         The authors just reported the ultimate load and moment for the experiments, while presenting the load-deflection or moment-deflection curve for the specimens can give further insight into the overall behavior of the members.

·         There are considerable deviations in Fig. 5 between FEM and Experiments; this should be explained.  

·          The details of FE simulations are missing and should be added.

Reviewer 3 Report

Authors are appreciated for proposing a new method of the design approach.

The following observations can be helpful in better understanding of the article.

Discussions for Fig 6, and Fig 7 is missing and it is a mere description of figures only.

The range of concrete compressive strain in fig 9 can be reduced to discuss in detail.

"The finite element analysis bending moment and their theoretical and experimental bending moment are very close to each other, which proves the correctness of the calculation method in this paper." looks a more generic statement and not as observed in the figure according to magnitude.

Authors may mention the material model used in the Finite element analysis considered in the article.

Reviewer 4 Report

The submitted research paper “materials-1886579” entitled: “The design of concrete beam reinforced with GFRP bars based on crack width” is mainly an experimental - analytical study. The authors proposed a new design method for GFRP-RC beams based on the flexure crack width. The authors have designed and experimentally tested 8 full-scale GFRP-RC beams under four point bending. The longitudinal as well as the transversal reinforcement of the beams consisted only of GFRP bars. The authors validated their proposed model by comparing with the experimental results. The authors also performed a numerical analysis and also compared the results to the proposed model and the experimental results.

The manuscript has some serious flaws in the research design method. Although the experimental program is consisted of sufficient number of experiments, the experiments are not presented in detail. Also the finite element analysis the way it is performed and presented does not offer anything further to the manuscript and could be interpreted as misleading or a try to fill up more pages in the manuscript. The authors must revise their manuscript thourougly, addressing the following comments would contribute to improvement of the manuscript:

1.     First, of all the introduction must be enriched by adding more recent studies in this very popular and evolving field. The literature review used through the manuscript is very limited and most of the studies mentioned are very old. The authors do not mention any study performed in the last 5 years and they only mention 2 studied from 2017 all the other studies are older. More recent studies need to be added so that the authors can prove the soundness of their study as well as what new does it bring to the scientific field. To help the authors I propose to read and discuss the following recent studies which will also help find others in this direction in the literature.

·       Flexural strength prediction for concrete beams reinforced with FRP bars using gene expression programming. Structures 2021 33, 3163–3172

·       Numerical Analysis Exterior RC Beam-Column Joints with CFRP Bars as Beam’s Tensional Reinforcement under Cyclic Reversal Deformations. Appl. Sci. 202212, 7419

·       Cyclic behavior of high-strength lightweight concrete exterior beam-column connections reinforced with GFRP. Buildings 202212, 179.

2.     The experimental program must be presented in more detail along with the details of all instrumentation used and the cast procedure. Providing figures with real photos of the setup and instrumentation and not only schemes would add to the manuscript.

3.     Photos and comparison of all experiments must also be provided with zoomed in areas of cracks so that the similarities and differences between the specimens can be observed and discussed.

4.     The authors must explain why did they add the finite element analysis. What does this analysis has to offer? Did they implement their proposed model in the finite element software? And if they did it must be thoroughly explained how. Finite element analysis are performed when a researched needs to examine more cases of an already verified FE model by running parametric analysis or when a FE model is proposed and has to be validated to experimental results. Which is the case in this study?

5.     If the authors want to present the FEA in their manuscript, all of the following must be described and added:

·       On what bases where the element types used for concrete and reinforcement selected? Why did they use truss elements for the GFRP bars?

·       What input parameters were used to simulate the bond between concrete and reinforcement surface? Why was the spring-element selected? What characteristics were given to the spring.

·       How many simulations did the authors run? Optical results of the simulations must also be provided.

·        Which constitutive laws were used for concrete and for GFRP?

·       Which was the mesh size and how was the mesh size selected? Was there any parametric analysis caried out?

·       Was the FE model validated? In figure 5 the FEA results are too far away from the corresponding experimental after the 1% of reinforcement ratio, it seems that the authors have not considered well the materials non-linearities.

·       The authors should also provide some images of the simulated beams in the software and as well some images with the results from the software where perhaps showing the distribution of stresses or damage and deformations and crack widths and comparing those figures to the experimental figures.

Round 2

Reviewer 4 Report

The authors have improved their manuscript and addresed the commments by making alterations or additions. The manuscript is now suitable for publication.